# Iterative Interactive Modeling for Knotting Plastic Bags

**Chongkai Gao**
Department of Automation
Tsinghua University
gck20@mails.tsinghua.edu.cn

**Zekun Li**
Department of Automation
Tsinghua University
lizk21@mails.tsinghua.edu.cn

**Haichuan Gao**
Department of Automation
Tsinghua University
ghc18@mails.tsinghua.edu.cn

**Feng Chen**
Department of Automation, Tsinghua University
LSBDPA Beijing Key Laboratory
chenfeng@mail.tsinghua.edu.cn

**Abstract:** Deformable object manipulation has great research significance for the robotic community and numerous applications in daily life. In this work, we study how to knot plastic bags that are randomly dropped from the air with a dual arm robot and image input. The complex initial configuration and intricate physical properties of plastic bags pose challenges for reliable perception and planning. Directly knotting it from random initial states is difficult. In this work, we propose Iterative Interactive Modeling (IIM) to first adjust the plastic bag to a standing pose with imitation learning to establish a high-confidence keypoint skeleton model, then perform a set of learned motion primitives to knot it. We leverage spatial action maps to accomplish the iterative pick-and-place action and a graph convolutional network to evaluate the adjusted pose during the IIM process. In experiments, we achieve an 85.0% success rate in knotting 4 different plastic bags including one that has no demonstration.

**Keywords:** Plastic Bag Manipulation, Learning from Demonstrations

## 1 Introduction

Deformable object manipulation (DOM) has been a long standing problem in robotics. Researchers have been studying manipulating various kinds of deformable objects, from linear objects [1, 2], fabrics [3, 4], papers [5] to elastic and elasto-plastic objects [6, 7, 8]. Apart from them, plastic bag, perhaps the most widely used application of plastics in our daily life, has remained unexplored in robotics literature due to the high complexity involved in modeling and controlling its deformation. Endowing robots with the ability to manipulate plastic bags can spawn diverse industrial and domestic applications in warehouses, garbage dumps, and supermarkets. Knotting plastic bags is one of the most representative manipulation tasks on plastic bags for a robot to show its dexterity in the real world. In this paper, we study how to knot plastic bags that are randomly dropped from the air with a dual arm robot and raw image inputs.

A randomly dropped plastic bag is shown in Figure 1(a). Knotting it from such an irregular initial configuration requires locating and taking the handles out from the mess and delicate coordination

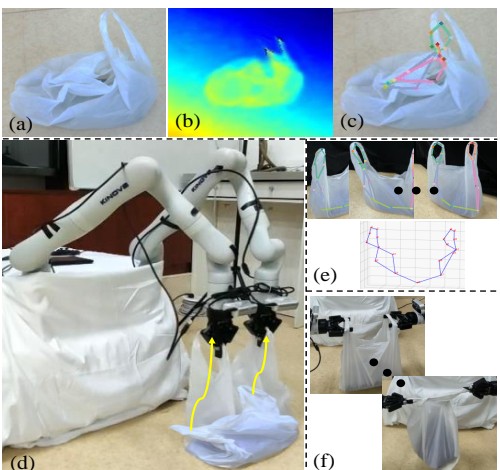

Figure 1: Overview of our work. (a) A randomly dropped plastic bag. (b) Inaccurate depth measurement from Intel Realsense D435i. (c) Unstable keypoint detection at the initial state. (d)(e) We propose IIM to first iteratively interact with the plastic bag to shape it to a standing pose, then build a high-confidence 3D keypoint skeleton with triangulation. (f) Knot tying with learned motion primitives.

6th Conference on Robot Learning (CoRL 2022), Auckland, New Zealand.

of dual arms to tie the knot. Compared to previous knot tying tasks that have been well studied on ropes [1, 9, 10, 11], this task poses new challenges for both perception and planning, because of the complex initial configuration and the intricate material and dynamics property of plastic bags. For perception, as shown in Figure 1(b), the translucent and non-Lambertian surface of plastic bags makes depth sensing inaccurate, thus 3D vision based models such as point cloud [12, 13, 14] or voxels [15] are not suitable for our task. Meanwhile, 2D vision models such as 2D keypoints [16] are not reliable due to the severe self-occlusion and partial observation problem caused by complex initial configurations (Figure 1(c)). For planning, mathematical physical models are difficult to build for plastic bags since their physical parameters are very hard to estimate, and to the best of our knowledge, no simulator can be employed. Learning a dynamics model for planning [7, 13, 15] is also infeasible since motions of plastic bags are not quasi-static. Although some image-based imitation or reinforcement learning approaches [3, 4, 17, 18] can directly learn a visual policy without a visual representation or physical model, their learned policies either employ just a small action space such as pick-and-place that is not enough to accomplish the knot tying task, or are unable to cope with complex initial configurations that appear in a randomly dropped plastic bag. Therefore, directly knotting the plastic bag from initial configurations is hard for both perception and planning.

Fortunately, the elasto-plastic property of plastic bags gives us another way to accomplish the knot tying task. Elasto-plastic objects can be shaped to a predefined target state in equilibrium and unchanging. This is called deformation control [19], and previous works have studied how to learn a dynamics model [13] or estimate physical parameters [20] of elasto-plastic objects such as sponges, plasticine, or clay by interacting with them during this shaping process. However, for our task, this interactive shaping process is more valuable for reducing difficulties of perception and planning: shaping a plastic bag from a randomly dropped state to a standing pose (Figure 1(d)) is a process of straightening it, and the partial observation and self-occlusion problem can be gradually alleviated during this process. This allows us to build a more reliable visual model (such as keypoint skeleton, Figure 1(e)), and the knot tying task can be realized from the standing pose based on this model. This adjusting process can be accomplished with only iterative pick-and-place actions [17, 4, 21] using recent advances, thus we can tackle the original problem by developing a visual learning policy to straighten the plastic bag and then acquiring a reliable perception model to perform the down-streaming knot tying task.

In this work, we propose Iterative Interactive Modeling (IIM) for knotting plastic bags randomly dropped from the air with only image input. We train the robot to first shape the plastic bag to a standing pose with the help of demonstrations and then establish a keypoint skeleton model with multi-view stereo images to knot it with a set of learned adaptive motion primitives. Specifically, the robot iteratively performs different kinds of top-down pick-and-place actions on part of the plastic bag to outspread it, meanwhile, a task progress module evaluates if the pose is good enough for knot tying. We leverage spatial action maps [17, 18] to accomplish the pick-and-place action, and train a graph convolutional network as the task progress module with the same demonstrations to evaluate the keypoint skeleton during the adjusting process. To enable keypoint detecting on plastic bags, we provide the first 2D plastic bag keypoint dataset *PBPose* with 43,200 images to train an off-the-shelf 2D keypoint detection model RLE [22]. After the IIM process, we lift the plastic bag into air with geometrically constrained planning to knot the plastic bag with a set of motion primitives with trained action parameters from CNN. In our experiments, we achieve an 85.0% success rate in tying four different plastic bags (one of them has no demonstration) that randomly dropped from the air with our dual Kinova Gen3 arms equipped with standard Robotiq 2F-85 grippers, with only 100 demonstrations (1.5 hours) provided for each plastic bag. In summary:

- We propose Iterative Interactive Modeling (IIM) for complex elasto-plastic object manipulation that iteratively shapes the object to facilitate more reliable perception and planning.

- We leverage spatial action maps, graph convolutional networks and the RLE model to perform IIM on plastic bags, and train motion primitives to accomplish the knot tying task.

- We build the first dual-arm robotic system to knot plastic bags randomly dropped from the air with the provided *PBPose* dataset and a small number of demonstrations.

## 2 Related Works

Deformable object manipulation (DOM) has long been a challenging area of robotics research. The challenges of DOM come from two properties of deformable objects: the infinite degrees of freedom and the complex non-linear dynamics, which lead to difficulties for perception, modeling, and plan-

ning. In this section, we compare different techniques for modeling and manipulating deformable objects and discuss whether they are suitable for knotting plastic bags.

## 2.1 Visual Representations of Deformable Objects

Keypoints [23, 24, 25] are commonly used representations. They are sparse representations for the structure of target objects, and grasping points are usually generated from them. Keypoint detectors can be trained either with supervised or unsupervised ways [26] to find keypoints with image or point cloud inputs to allow robots to manipulate the object based on some geometry loss on the keypoints. Dense visual descriptors [8, 27, 28, 29] are recently used as a real-time pixel-wise dense representation for many kinds of objects, and humans can specify grasping points on it. Besides, the 3D vision community has made great progress in modeling deformable objects, with voxels [30], meshes [31], convexes [32], or implicit functions such as flow-based model [33] or neural radiance fields [34], and most of these model can represent the deformable object in high fidelity. Other works seek to use particles [35, 14, 36] that are down-sampled from point cloud data to represent deformable objects. These methods usually combine graph neural networks and differentiable simulators [37, 38] to learn the dynamics of objects. Lastly, visual policy learning methods simply use high-dimensional feature embedding learned from deep neural networks as object representations for manipulation [3, 4, 11, 39] and have achieved great success in various tasks.

For plastic bags, current commercial depth sensors fail to give accurate depth estimation because of the translucent and non-Lambertian surface, thus 3D vision based methods are not feasible. Since we aim to manipulate the plastic bag rather than studying how to build a refined representation, those high-fidelity representations are not necessary: they all degenerate to grasping points [28] in practice for manipulation. Thus, we choose keypoint skeleton as the representation of plastic bags.

## 2.2 Manipulating Deformable Objects

A lot of work aims to build a dynamics model for the target object first and then perform motion planning on it to manipulate the object. Conventional works build the dynamics model mathematically [40, 41, 42], and some of them have achieved promising results recently by planning on simpler approximated dynamics and using local controllers to tackle actual complex dynamics [43, 44, 45]. Other works aim to learn the dynamics model from visual data [13, 15, 7] or tactile data [46, 47]. Recently, reinforcement learning and imitation learning methods are developed to directly manipulate the object without a dynamics model, and have achieved success on various tasks [3, 48, 1, 10, 17, 49]. Some of them train the robot in simulation with fast virtual experiences and then transfer the learned policy directly to real robots or with sim-to-real methods [27, 50]. Others try to directly train the robot in the real world with its own experience or expert demonstrations [4].

For manipulating plastic bags, dynamics models are hard to build since the complex non-linear dynamics, and the motion of plastic bag is not always quasi-static: some parts of the plastic bag may collapse after it has been moved. In this work, we follow the idea of most visual policy learning approaches that directly learn a pick-and-place policy with a limited action space to accomplish the adjusting task, and use geometrically constrained planning to lift the plastic bag to the air to knot it with a set of learned motion primitives.

## 3 Method

The goal of this paper is to enable a dual arm robotic system to knot plastic bags from initial randomly dropped states. We use keypoint skeleton as the visual representation for plastic bags. The robot is trained to iteratively adjust the plastic bag to a standing pose to build a complete and high-confidence keypoint model with the help of a task progress module, then tie the knot with a set of learned motion primitives. We introduce the problem setting in 3.1, the keypoint detection model in 3.2, the iterative interactive modeling process in 3.3, and the knotting process in 3.4.

## 3.1 Problem Statement

We use the most common type of plastic bags that are colored, medium-sized (33.7~42.5 cm wide and 40.2~42.3 cm long when laid out), translucent, and have two looped handles, as shown in Figure 2(a). We choose four kinds of plastic bags that vary in size and color. In order to make the tying problem possible, we put some items in the plastic bag to resist indoor airflow. The plastic bag is randomly dropped from the air to form an initial state. The knot we tie in this work is a kind of *Ian Knot* [9] which can be tied with two motion primitives (Action 1 and Action 2) from the stage of hanging on grippers, as illustrated in Figure 2(c): 1) The right arm grasps the rear side of the left

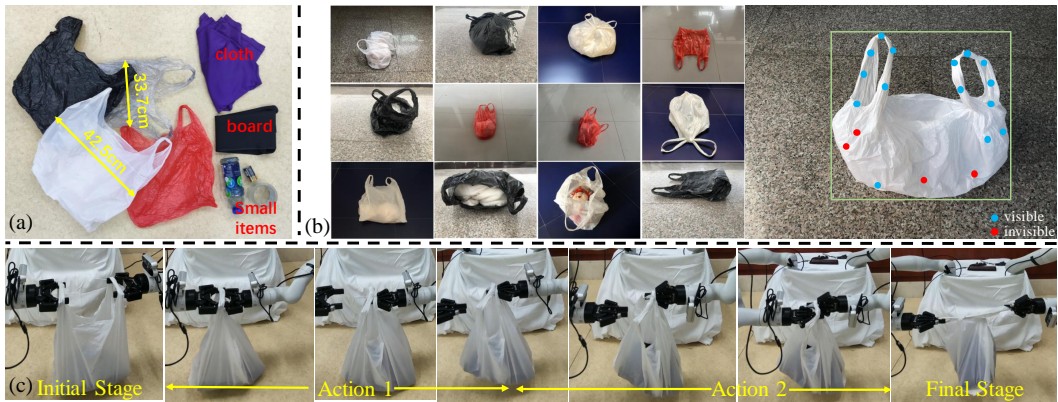

Figure 2: (a) Plastic bags and fillers used in this work. We give demonstrations on white, red and black bags. The grey plastic bag is used to show the generalization of IIM. We fill plastic bags with different items varying in size and shape, including cloths, paperboards, bottles, tape volumes, and dry batteries. (b) The *PBPose* dataset. Images are captured in various backgrounds, knotting states, occupied states, illumination conditions, placement status (standing or lying down), and angles of view. (c) We design two motion primitives for tying an *Ian Knot* in the air.

handle to allow the left arm to go back; 2) The right arm rotates $90°$ to let the left arm grasp the front side of the right handle. The knot is tightened by pulling two handles away from each other.

The whole experimental setup is shown in Figure 3. We use two identical Kinova Gen3 (6DoF) arms and Robotiq 2F-85 grippers for our experiments. We mount two Intel Realsense D435i cameras $C_{left}$ and $C_{right}$ on the end of each arm respectively to get image inputs $I_{left}$ and $I_{right}$ from the end-effector perspective. The distance between the bases of two arms is 51.2cm. $C_{right}$ looks straight down at the $XY$ plane from a top view to get $I_{right}$ within the range of 75.4cm × 56.6cm with the plastic bag at the center. On the other side, we get $I_{left}$ by moving $C_{left}$ aiming at the plastic from a set of widely-separated poses to get stereo images.

### 3.2 3D Keypoint Skeleton for Plastic Bags

Directly 3D keypoint detecting methods and reconstruction methods are not suitable for plastic bags as discussed in section 2.1. Thus in this work we detect 2D keypoints of plastic bags with image input and reconstruct 3D keypoint skeleton with multi-view geometry. This requires a plastic bag keypoint dataset and a well-trained 2D keypoint detection model.

To this end, we provide *PBPose*, the first 2D plastic bag keypoint dataset with 43,200 images of four different kinds of plastic bags. Images in *PBPose* vary from the background, knotting state, occupied state, illumination condition, placement status, and angle of view, as shown in Figure 2(b). For each image, we manually label 19 keypoints of the plastic bag from handle to bottom to establish a full skeleton of it, along with point visible properties and the bounding box. Based on this dataset, we train a RLE network [22] $f$ for 2D keypoint detection. $f$ takes as input of a single frame from $I_{left}$ and outputs predicted 2D joints of the target plastic bag along with confidences of each point.

From the illustrated performances on plastic bags at initial states and standing poses in Figure 1(c) and 1(e), we can see that it is not robust and reliable to detect keypoints from random initial configurations. Although this problem can be mitigated by providing more training data at different initial configurations, it is hard to cover enough cases of the infinite continuous initial states. Thus in this work we sidestep this problem by shaping the plastic bag to a standing pose.

### 3.3 Iterative Interactive Modeling

We adjust the plastic bag from its initial state by iteratively picking and placing it with expert demonstrations, and automatically evaluate the pose with a task progress module. We introduce these modules one by one in the following part. Note we do not require the whole adjusting process to be quasi-static, which means the plastic bag can partially collapse before it can steadily stand.

**Observation and Action Space Definition**: At every step, we get $h$ multi-view images $I_{left}^{mv} = \{I_{left}^0, \cdots, I_{left}^{h-1}\}$ from $C_{left}$ ($h = 15$ in this work), and one top-view image $I_{right}$ from $C_{right}$. The robot can choose two kinds of pick-and-place action at each step: 1) a single-arm picking action

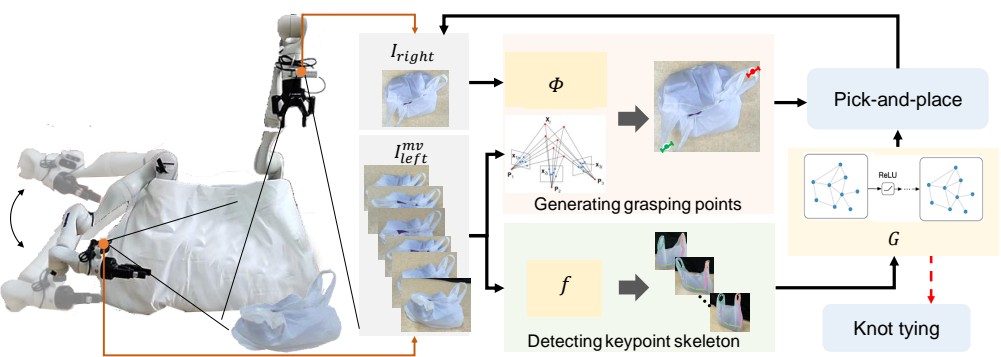

Figure 3: The IIM iteration for knotting plastic bags. At each step, the right camera gets a top-down view image $I_{right}$, and the left camera gets a sequence of side view images $I_{left}^{mv}$. We use a spatial action maps module $\phi$ to get the grasping points, and use a sparse reconstructed point cloud to get the grasping depth. At the same time, we detect 2D keypoints from $I_{left}^{mv}$ with $f$ to facilitate the task progress module $G$ to determine which picking action to use, and when the adjusting pose is good enough to support building a 3D keypoint skeleton to tie the knot.

$p_1$ that goes top-down to location $R \in \mathbb{R}^3$ and picks up to 0.55m height along the $Z$ axis; 2) a dual-arm picking action $p_2$ that goes top-down to locations $L, R \in \mathbb{R}^3$ and picks up to 0.55m height along the $Z$ axis respectively. After each picking, the arm can choose to go to a left/staying/right placing position to release the plastic bag, as shown in Figure 4. $p_1$ is used for most pick-and-place actions and $p_2$ is only used for final steps to get a good standing pose.

The robot needs to choose which action to use, and grasping and placing locations at every step. Specifically, the $xy$ locations of grasping points are determined by our learned visual policy. The $z$ parameter (grasping depth) is determined by first running sparse reconstruction of stereo images by structure from motion algorithm (SfM) to get sparse point cloud (using COLMAP [51] in this work), then calculating the average height of the top-10 points on a cylinder with a radius of $1cm$ with the center of the bottom surface of the grasping point to get the grasping depth. For determining the placing location, we segment $I_{right}$ and calculate the relative sizes of the left-side and right-side areas. If the ratio of two areas is less than 0.5 or greater than 2, the robot will place the plastic bag to the smaller-side placing point. Otherwise it will release the gripper in situ, as shown in Figure 4.

**Demonstration Collection**: A human expert demonstrates how to accomplish the adjusting process for the robot. For each demonstration trajectory $D = \{p_{c_0}, p_{c_1}, \cdots, p_{c_{k-1}}\}$, it includes $k$ picking actions, where $c_0, c_1, \cdots, c_{k-1}$ denote picking categories. For each $p_{c_i}$, we record $I_{left}^{mv}$ and $I_{right}$ before performing the action, and the grasping locations $L, R \in \mathbb{R}^2$ from the perspective of $C_{right}$, along with the grasping angles $\theta_L, \theta_R$. Thus, $p_{c_i} = \{I_{left}^{mv}, I_{right}, L, R, \theta_L, \theta_R\}_{c_i}$. The expert always grasps *handles* of the plastic bag for all picking actions. This is important to establish a unified grasping principle to avoid ambiguity for imitation learning. $p_2$ may be used several times at the end of a trajectory to form a good standing pose. In this paper, we find 100 demonstration trajectories are enough for each kind of plastic bag, which only take 1.5 hour for human to demonstrate.

**Visual Grasping Module**: The robot determines the grasping points $\{(L, \theta_L), (R, \theta_R)\}$ from $I_{right}$ with spatial action maps [17, 21, 18], which have shown promising results for learning heatmaps of visual-affordances over pixels with fully convolutional networks. They can recover pixel-wise grasping affordances of different graspings with input images being rotated to achieve a discrete set of possible actions. Concretely, as shown in Figure 4, given an image from $I_{right}$, we first segment out the plastic bag from the background and generate 16 rotated images (multiples of 22.5°), then pass them through the fully-convolutional network $\phi$ to predict the corresponding set of heatmaps within the same size of the input image. In this work we use a pixel resolution of $160 \times 120$. The pixel with a higher predicted probability among all 16 maps is more suitable for grasping.

The problem in our case is that we have to determine one grasping point for $p_1$ but two grasping points for $p_2$. To this end, we propose a *Selection Rule* based on safe distance constraints to avoid collisions of grippers: for $p_1$ and $p_2$, the pixel with the highest value corresponds to $(R, \theta_R)$. When choosing $(L, \theta_L)$ for $p_2$, we first delete nearby areas (a circle of 5 pixels radius) of $(R, \theta_R)$ on all 16

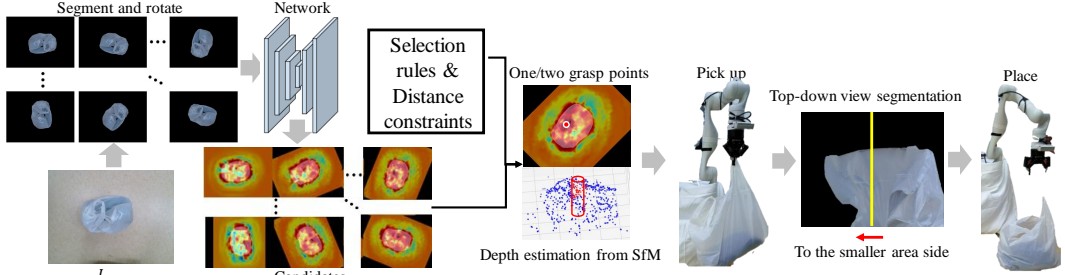

Figure 4: A pick-and-place process. $I_{right}$ is segmented and rotated to get 16 copies. We send these images through network $\phi$ and choose one or two grasping point(s) from the output candidates according to our selection rules and distance constraints. We get the grasping depth of these points by calculating the depths of top-10 points above the grasping points and perform the picking action. After picking it upward, we determine where to place the plastic bag by segmenting the top-down view and compare relative sizes of left-side and right-side area of the plastic bag.

output images, and select the pixel with the highest value in the remaining area. If this pixel has a value higher than 0.8, then we choose it as $(L, \theta_L)$, otherwise we abort $p_2$ and only perform $p_1$.

**Task Progress Module**: To evaluate the quality of poses during the adjusting process to determine when the robot can use $p_2$ and when the pose is good enough to facilitate knot tying, we propose to use a graph convolutional network to classify the keypoint skeleton during the IIM process to predict the *task progress* stages. We represent the 2D keypoint skeleton as a graph formed by state $\mathbf{S} = (\mathcal{O}, \mathcal{E})$, where vertices $\mathcal{O}$ are the keypoints. Concretely, $\mathbf{o}_i = \langle \mathbf{x}_i, \mathbf{c}_i \rangle$, where $\mathbf{x}_i$ and $\mathbf{c}_i$ are 2D pixel locations and confidences of each keypoint respectively. The edges $\mathcal{E}$ are the connectivity of each pair of keypoints that are predefined. At each step, a graph convolutional network $G$ gets $h$ graphs abstracted from stereo image inputs of $I_{left}^{mv}$ and detected keypoint skeletons by $f$ and calculates the mean state $\bar{\mathbf{S}}$. Then it classifies current step to three categories: 1) ordinary picking step (using $p_1$); 2) final picking step (using $p_2$); 3) ending step (no picking action will be chosen). We train $G$ with the standard cross entropy loss and the same demonstrations collected above.

After IIM, we get a well-standing plastic bag. We recover the 3D keypoint skeleton of it by performing triangulation from detected 2D keypoints of stereo images of $I_{left}^{mv}$ with the help of calibrated camera intrinsic and extrinsic parameters. The next step is to knot the plastic bag with this model.

### 3.4 In-air Knotting Plastic Bags by Learning Motion Primitives

As shown in Figure 2(c), tying an Ian Knot needs to first lift the plastic bag to the air to eliminate the influence from ground. This requires the direction that allows each arm to go into each ring on handles, as shown in Figure 5(a). There are six keypoints on each ring, and they are not on the same plane in $\mathbb{R}^3$. Thus we here calculate a *ring plane* that minimizes the total least squares (TLS) distance from each point to it. Then we calculate the perpendicular direction of each ring plane, and let this perpendicular go through the center point of the ring, which is the average of 3D coordinates of all the ring points. By TLS, we know this center point is on the ring plane too. We set each robot arm

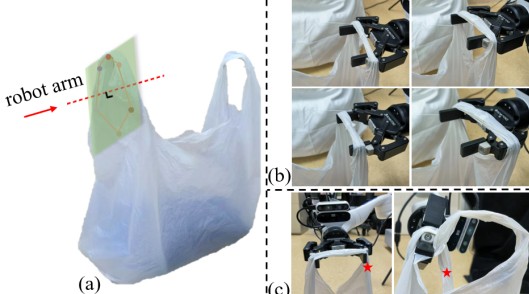

Figure 5: (a) Calculating the inserting direction. (b) Different initial stages for tying Ian Knot. (c) End-effector view before action 1 and action 2. Red stars represent goals.

on its perpendicular respectively and place each gripper $0.1m$ away from the center point of each ring. Then we close both grippers and move the arm along the perpendicular to insert into the ring, and keep both end effectors horizontal with a 0.25m distance to lift the bag to a height of 0.5m. Finally we open both grippers to reach the *initial stage* for tying an Ian Knot.

The second step is to perform the two motion primitives. However, as shown in Figure 5(b), the dangling handles of plastic bags on the gripper may hang in different specific places. This requires both actions to adaptively choose their goals (the grasping points, see Figure 5(c)) according to

Table 1: Success rates of different methods. FSR of baseline methods are calculated by using our method to complete the missing parts of them.

| Metircs | Plastic Bags | DI | RP | VBE-S | VBE-T | HKT | Ours |
|---|---|---|---|---|---|---|---|
| ASR | Red, Black, White | 4.0%(2/60) | 13.3%(8/60) | 70.0%(42/60) | 65.0%(39/60) | - | **88.3%(53/60)** |
| | Grey (Unlearned) | 5.0%(1/20) | 10.0%(2/20) | 40.0%(8/20) | 30.0%(6/20) | - | **80.0%(16/20)** |
| KSR | Red, Black, White | - | - | - | - | 41.6%(25/60) | **98.3%(59/60)** |
| | Grey (Unlearned) | - | - | - | - | 40.0%(8/20) | **90.0%(18/20)** |
| FSR | Red, Black, White | 4.0%(2/60) | 13.3%(8/60) | 70.0%(42/60) | 63.3%(38/60) | 38.3%(23/60) | **86.7%(52/60)** |
| | Grey (Unlearned) | 0.0%(0/20) | 10.0%(2/20) | 30.0%(6/20) | 25.0%(5/20) | 25.0%(5/20) | **80.0%(16/20)** |

actual situations. To this end, we train a 4-layer convolutional neural network using Huber loss with hidden size=256 to regress grasping points for both actions directly in $\mathbb{R}^3$ with images input, as shown in Figure 5(c). Following up demonstrations in 3.3, we extend each demonstration with a knot tying part. For each demonstration, we record two images from right and left end-effector views respectively for action 1 and action 2, and record corresponding grasping locations in $\mathbb{R}^3$.

# 4 Experimental Results

## 4.1 Metrics and Baselines

We evaluate our method on success rates of knot tying on three plastic bags that have demonstrations on them and a new type of plastic bag (grey) that has no demonstration on it to show the generalization ability of our method. Concretely, we evaluate: a) Adjusting Success Rate (**ASR**): if the plastic bag is successfully adjusted to a standing pose from a randomly dropped initial state to facilitate a valid inserting and lifting action (evaluated by human); b) Knot Tying Success Rate (**KSR**): if the plastic bag is successfully knotted from a randomly in-air initial stage. c) Full Task Success Rate (**FSR**): if the plastic bag is successfully knotted from a randomly dropped initial state. For ASR and FSR, we say one attempt fails if it does not succeed in 10 steps of grasping. We show the effectiveness of different modules in our method by a set of ablated versions:

**Directly Inserting without Adjusting (DI):** This method aims to directly find the handles of the plastic bag to lift it with the same method in 3.4 when it is just dropped from the air by the keypoint skeleton detected at the initial state. This is used to show the necessity of IIM.

**Random Picking (RP):** The robot randomly picks up a point of the plastic bag based on the segmented plastic bag area from top-down view image $I_{right}$ and randomly generated a single grasping point in this area. This is used to show the effectiveness of the spatial action maps $\phi$.

**Vision Based Keypoint Skeleton Evaluation (VBE):** This method evaluates the goodness of the adjusted pose of the plastic bag directly from images rather than using a GCN to process the keypoint skeleton. This is used to show the effectiveness of the task progress module $G$. We use channel-wise concatenated side view images (**VBE-S**) and top-down view images (**VBE-T**) for classifying stages in this baseline respectively.

**Hard-coded Knot Tying (HKT):** This method ties the Ian Knot with the same primitives of our method but use hard-coded goals for each action. This is used to show the effectiveness of learned motion primitives by CNN.

## 4.2 Quantitative Results

Table 1 shows the ASR, KSR, and FSR of different baselines. Our method achieves the best success rates in all metrics. For ASR, DI can barely knot the plastic bag, since most of the initial states do not support high-confidence keypoint detection (as shown in Figure 6), but occasionally a randomly dropped plastic bag can form a good standing pose. RP achieves similar results. Random picking can stretch out the plastic bag to some extent, but it can never end up with a good standing pose, which needs a dual-arm picking action $p_2$. VBE methods achieve much better results than the above two methods on the training plastic bags, especially for VBE-S. However, their performance drops dramatically on the grey plastic bag. This is because although images can also provide information about poses of plastic bags, they do not extract and use the essential information for evaluating poses such as the keypoint skeleton, and are susceptible to different illumination conditions. Meanwhile we only provide hundreds of images for each plastic bag, which may not be sufficient for image-based classification. That is why VBE methods are not as good as IIM on training plastic bags and lack generalization abilities to new plastic bags. For KSR, the hard-coded knot tying method can achieve an average 41.25% success rate. Most of failures of HKT happen in action 2. The goal position in action 2 varies because it is affected by the grasping results of action 1. A hard coded action 2 will miss the front side of the right handle or just grasp a wrong part of the plastic bag.

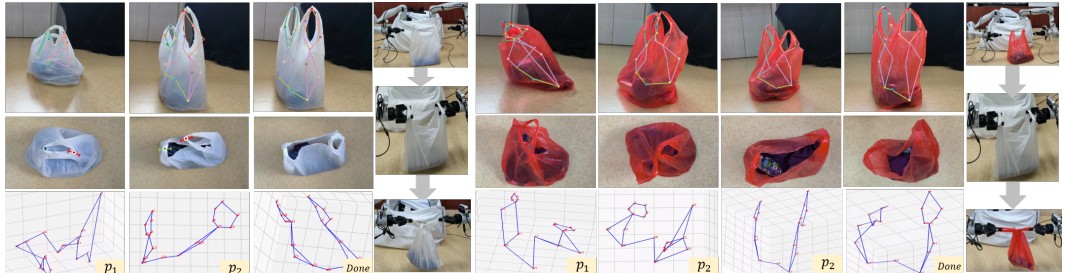

Figure 6: IIM processes in white and red plastic bags. Top row: side view and detected keypoint skeleton of plastic bags at each step. Middle row: top-down view and the predicted grasping points and directions. Bottom row: reconstructed 3D keypoint skeletons and the predicted actions at each step. We can see at both initial states, handles interleave with each other, but the robot can finally adjust them to standing poses with iterative pick-and-place actions.

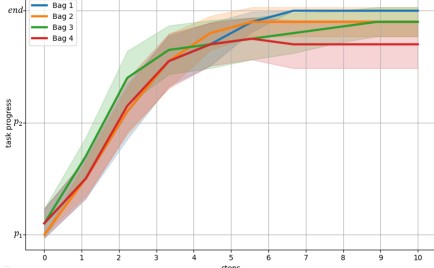

Figure 7: All task progresses along with grasping steps.

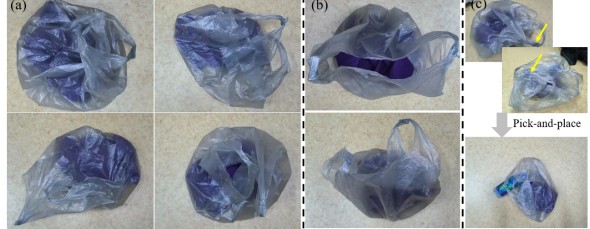

Figure 8: (a) Different initial configurations. (b) Special initial configurations that support directly lifting. (c) Failure cases. The filler is thrown out.

### 4.3 Qualitative Results

We show the qualitative results of white and red plastic bags in Figure 6 and all task progress curves evaluated by $G$ (mean and std) in Figure 7. IIM shows great generalization ability to adjust plastic bags from different initial states to the standing poses. Most IIM processes finish in 4 steps. Some of them have more steps to finish because of harder initial configurations: one handle is too closer to the surface below it, or two handles interweave. In this situation the robot may grasp up more parts than one handle, but after several pick-and-place, the interlaced part can disentangle from each other and subsequent graspings become normal, as shown in the red plastic bag in Figure 6.

**Failure Cases:** The inaccuracy of depth estimation is the main reason that causes a failed grasping. Some graspings fail to catch the bag (above the right grasping point). Others exceed the right grasping depth. The first case will make the robot stuck in a loop. The second case will make the robot grasp the body of the plastic bag directly. This may lead to a violent jolt when placing the plastic bag, which almost equals starting all over again. Sometimes the filler items fall out, which leads to a failed grasping, as shown in Figure 8(c). Most of the attempts can finish in 5 grasps, but some attempts (around 15%) need more steps due to the above reason, and 13.3% of attempts fail.

## 5 Discussion and Future Works

In this work we propose Iterative Interactive Modeling for knotting plastic bags that are randomly dropped from the air with a dual arm robot. We show the effectiveness of various visual learning methods such as spatial-action maps and keypoint detection models on plastic bags. IIM is a general type of interactive perception [52] for modeling complex elasto-plastic objects: interacting with objects to complete an explicit representation model. Modules in our method can be replaced by other specific methods for different objects. For example, the representation model (keypoint skeleton in this work) can be dense descriptors [8, 27, 28] or partical-based graphs [13, 14, 36], and the completion algorithm (imitation learning in this work) can be graph-based completion algorithms.

The limitations of our work are: 1) The keypoint detector depends on our custom dataset *PBPose*, and IIM is based on human demonstrations. These make our method lack quick extensibility to other objects and situations. 2) The assumption of this work is that the plastic bag can stand still after being adjusted. For some very soft/hard plastic bags, or some fillers that do not support standing, our method would not be effective. 3) Our method cannot handle extremely difficult initial configurations such as the handles being pressed underneath the plastic bag. Future works may seek to employ more actions such as pushing to tackle these situations.

**Acknowledgments**

We would like to thank Tianren Zhang and Yizhou Jiang for their insightful comments of the whole work, and Qualcomm China WRD UR Program. This work was supported by the National Natural Science Foundation of China 62176133.

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
