# OpenReview forum: "Iterative Interactive Modeling for Knotting Plastic Bags"
_robot-learning.org/CoRL/2022/Conference — CoRL 2022 Poster_

### Official Review · Reviewer_6XFW · 2022-07-29

**Originality:** Very Good
**Technical Quality:** Very Good
**Clarity Of Presentation:** Excellent
**Impact:** 3

**Recommendation:**

Strong Accept: I recommend accepting the paper and will argue for my recommendation even if other reviewers hold a different opinion.

**Summary:**

The authors proposed the Iterative Interactive Modeling (IIM) to first adjust the plastic bag to a standing pose with imitation learning to establish a high-confidence key point skeleton model. Directly knotting plastic bags from random initial states is difficult because of the complex initial configuration and intricate material and dynamics properties of plastic bags. To solve this problem, the authors used keypoint skeletons to obtain a visual representation from the complex shape of the plastic bag. The robot is trained to iteratively adjust the plastic bag to a standing pose to build a complete and high confidence keypoint model, then tie the knot with a set of learned motion primitives. The authors demonstrated that a plastic bag dropped randomly from the air can be tied together by a dual-armed robot using the proposed method.

**Issues:**

The paper was clearly written and the reviewer pointed out only two minor corrections wrote in "Strengths And Weaknesses".

**Quality Of The Limitations Section:**

Limitations are addressed clearly

**Reviewer Expertise:**

4: The reviewer is confident but not absolutely certain that the evaluation is correct

**Robotics Focus:**

Sufficient demonstration on hardware

**Strengths And Weaknesses:**

The authors robustly estimate the grasping point of a plastic bag, which is difficult to recognize, and two additional motion primitives enable Ian Knot in the air with a dual-armed robot. Despite being a very challenging task, it is wonderful to show that the task can be performed with high accuracy with only 100 demonstrations. The supplementary video and source code attached were helpful to the reader's understanding. After clearly stating the difficulty of recognizing plastic bags directly from 3D keypoints in a related work, the authors propose a method for solving the problem. The authors propose a method for generating 3D keypoints using 2D keypoints and multi-view geometry. The specific model structure is referenced in the source code, and the main body of the text describes the data collection conditions and training methods, allowing the reader to conduct additional experiments. Chapters 3.3 and 3.4 were similarly clearly written with issues and methods that the reviewer were able to understand well. The evaluation experiment describes the success rate of the task based on three evaluation metrics using four different comparison methods. In addition, the authors mention generalization performance using untrained plastic bags and failure cases, and the reviewers consider the experimental results are sufficient.

Minor comments are as follows.
"A human expert demonstrates how to accomplish the adjusting process," but what exactly is being demonstrated?
How long does the task take to execute? It would be good to add the execution time to the text and display the playback speed (e.g. 4x) of the video.

**Summary Of Recommendation:**

The authors' research is high quality because they have performed a challenging task with high accuracy, and furthermore, the proposed method is novel and well described.

---

> ### Author Response · Authors · 2022-08-19
> **Response to Reviewer 6XFW**
>
> Dear reviewer 6XFW,
>
> We sincerely appreciate your valuable and insightful comments. We address your questions in detail below.
>
>
> ### Q1: What exactly is being demonstrated to accomplish the adjusting process?
>
> A1: In this work, a human expert demonstrates how to pick up the plastic bag. Specifically, for one data point in the demonstration, the expert takes a top-down view image $I_{right}$, labels the grasping locations $\mathcal{L}, \mathcal{R}$ and angles $\theta_{L}, \theta_{R}$ on the image, then picks up the plastic bag by grasping the given grasping locations and angles.
>
> ### Q2: How long does the task take to execute? What is the playback speed of the video?
>
> A2: Thanks for this question and your suggestion. How long the entire task takes depends on how many times the top-down pick-and-place actions are used. One pick-and-place will take around 30 seconds (including taking a top-down image, taking side-view images, and performing top-down grasping), and the knot tying process takes around 25 seconds. Thus, take the most common case (4 times of top-down grasping) as an example, the whole task will take 30*4+25=145 seconds to complete. The playback speed is 5x.

---

### Official Review · Reviewer_eMeZ · 2022-07-31

**Originality:** Good
**Technical Quality:** Very Good
**Clarity Of Presentation:** Very Good
**Impact:** 4

**Recommendation:**

Weak Accept: I recommend accepting the paper, but will not argue for my recommendation if the majority of other reviewers have a different opinion.

**Summary:**

The paper is about how to knot plastic bags. The high level idea is that estimating the state of plastic may not be possible at all times as a result the task is decomposed in to straightening up the bag first until the keypoint estimation becomes good and then solve the task based on the predicted keypoints on the plastic bag. The method is relying on a task progress module to decide which model to use and which phase of the task the robot is. The task progress module uses graph convolution network on the predicted keypoints along with their confidences to determine whether the robot needs to straighten up the plastic bags, approach the handles for knotting, and do the actual knotting. The model for straightening the bags is basically learned from demonstration data using action maps. For the final knotting, 2 planes are fitted from keypoints detected on each bag handle and that normal of that plane will be the approach direction of arms.


**Issues:**

See weaknesses above.

**Quality Of The Limitations Section:**

Limitations are addressed clearly

**Reviewer Expertise:**

5: The reviewer is absolutely certain that the evaluation is correct and very familiar with the relevant literature

**Robotics Focus:**

Sufficient demonstration on hardware

**Strengths And Weaknesses:**

Strengths:
The task is important and challenging to tackle.
Experiments are done on a dual-arm manipulator.

Weaknesses:
IMO, the method needs to be re-ordered. I think it flows better if it starts from task progress module and then go over all the other modules.
I am not super clear on the actual knotting process. Is it open loop? It seems like the 3D keypoint detections are trained with only non-interactive data. Also, wrist mounted cameras basically don’t see much at close distance
It seems like learning from demonstrations work decently well for straightening up the plastic bug. Why not do it for the approach and knotting process? Why is the keypoint detection crucial for this task?
The method seems extremely engineered for this specific task.


**Summary Of Recommendation:**

I think overall the problem is challenging and this paper will at least present a compelling baseline for future works to build on top of.

---

> ### Author Response · Authors · 2022-08-19
> **Response to Reviewer eMeZ (2/2)**
>
> ------
> [1] K. Suzuki, M. Kanamura, Y. Suga, H. Mori, and T. Ogata. In-air knotting of rope using dual arm robot based on deep learning. In 2021 IEEE/RSJ International Conference on Intelligent Robots and Systems (IROS), pages 6724–6731. IEEE, 2021.
>
> [2] Y. Wu, W. Yan, T. Kurutach, L. Pinto, and P. Abbeel. Learning to manipulate deformable objects without demonstrations. arXiv preprint arXiv:1910.13439, 2019.
>
> [3] A.Wang, T. Kurutach, K. Liu, P. Abbeel, and A. Tamar. Learning robotic manipulation through visual planning and acting. arXiv preprint arXiv:1905.04411, 2019.
>
> [4] P. R. Florence, L. Manuelli, and R. Tedrake. Dense object nets: Learning dense visual object descriptors by and for robotic manipulation. arXiv preprint arXiv:1806.08756, 2018.
>
> [5] A. Ganapathi, P. Sundaresan, B. Thananjeyan, A. Balakrishna, D. Seita, J. Grannen,M. Hwang, R. Hoque, J. E. Gonzalez, N. Jamali, et al. Learning dense visual correspondences in simulation to smooth and fold real fabrics. In 2021 IEEE International Conference on Robotics and Automation (ICRA), pages 11515–11522. IEEE, 2021.
>
> [6] P. Florence, L.Manuelli, and R. Tedrake. Self-supervised correspondence in visuomotor policy learning. IEEE Robotics and Automation Letters, 5(2):492–499, 2019.
>
> [7] H. Shi, H. Xu, Z. Huang, Y. Li, and J. Wu. Robocraft: Learning to see, simulate, and shape elasto-plastic objects with graph networks. arXiv preprint arXiv:2205.02909, 2022.
>
> [8] Y. Li, J. Wu, R. Tedrake, J. B. Tenenbaum, and A. Torralba. Learning particle dynamics for manipulating rigid bodies, deformable objects, and fluids. arXiv preprint arXiv:1810.01566, 2018.
>
> [9] Y. Li, T. Lin, K. Yi, 438 D. Bear, D. Yamins, J. Wu, J. Tenenbaum, and A. Torralba. Visual grounding of learned physical models. In International conference on machine learning, pages 5927–5936. PMLR, 2020.

---

> ### Author Response · Authors · 2022-08-19
> **Response to Reviewer eMeZ (1/2)**
>
> Dear reviewer eMeZ,
>
> We sincerely appreciate your valuable and insightful comments. We address your questions in detail below.
>
> ### Q1: I think it flows better if it starts from task progress module and then goes over all the other modules.
>
> A1: Thanks for your suggestion. Yes, the task progress module is kind of a *brain* of the whole process. It is feasible to start telling the story from the task progress module, which will make our paper become a top-down narrative. However, our current way to tell the story starts from the keypoint skeleton model, from which we come up with the idea to shape a good posture for plastic bags to get reliable keypoint detection. This makes our current story a bottom-up style. We think both ways have their own advantages: starting from the keypoint skeleton will inspire more practical technique thinking, and starting from the task progress module will inspire more philosophical thinking.
>
> ### Q2: Is the actual knotting process open loop?
>
> A2: Yes, it is an open loop. Once the wrist-mounted camera determines the grasping point, the robot arm will execute its command. Lots of recent works on tying knots are also open loop [1][2][3].
>
> ### Q3: Wrist mounted cameras basically don’t see much at close distance.
>
> A3: Yes, and that’s why we use two wrist-mounted cameras in this work to be far away from the plastic bags to get wide enough views. The right camera is responsible to get top-down view images, and the left camera is responsible to get side-view images. The plastic bag is placed a little bit to the right side to make the left camera has enough distance. In practice, the distance between the plastic bag and both cameras are all more than 55 cm, which is enough for a Realsense D435i camera to get a wide enough view.
>
> ### Q4: Why not do learning from demonstrations for the approach and knotting process?
>
> A4: Good question. We think that your meaning is “why not make all motions of robot arms during the knot tying process an imitation learning problem, rather than just learning the grasping locations?” This is because the knot tying process is too long for most robot imitation algorithms to learn a reliable policy with only a few demonstrations. Previous works [1][2] also just predict the grasping locations on ropes and call their methods learning-based methods. Thus, in this work, we follow their ideas to learn and predict the grasping points for different motion primitives.
>
>
> ### Q5: Why is the keypoint detection crucial for this task?
>
> A5: According to our discussion in the second paragraph in Introduction and the second paragraph in Section 2.1, we think that keypoints are appropriate visual representations for plastic bags in our task. Thus the core problem of this paper is to establish a reliable 3D keypoint skeleton.
>
> ### Q6: The method seems extremely engineered for this specific task.
>
> A6: Yes, up to a point, that’s true. However, here we want to emphasize that Iterative Interactive Modeling (IIM) is a general idea for manipulating objects in many challenging situations: interacting with objects to complete an explicit representation model and evaluating the current representation with an evaluating module, then performing then manipulation based on the representation. In our paper, we use keypoint skeleton as our representation, evaluate it with GCN, and complete it by iteratively pick-and-place with demonstrations. These modules can be replaced by other specific methods for different objects and tasks. For example, the representation model (keypoint skeleton in this work) can be dense descriptors [4][5][6] or particle-based graphs [7][8][9], and the completion algorithm (imitation learning in this work) can be reinforcement learning algorithms.

---

### Official Review · Reviewer_5bQ2 · 2022-07-31

**Originality:** Very Good
**Technical Quality:** Very Good
**Clarity Of Presentation:** Very Good
**Impact:** 4

**Recommendation:**

Strong Accept: I recommend accepting the paper and will argue for my recommendation even if other reviewers hold a different opinion.

**Summary:**

This paper tackles a novel and interesting task: knotting plastic bags. The proposed method, Iterative Interactive Modeling (IIM), contains keypoint detection, single/dual arm pick and place, and knotting primitive. Finally, the method is trained and evaluated in the real-world setup.

**Issues:**

- It will be better to include more details of each submodule. For example, how to train the spatial action map using demonstration.
- The dual arm pick-and -place policy may be potentially improved. For example, FlingBot uses a scaled and rotated spatial action map to predict the dual-arm picking policy.
- I just realized the picking policy is pretty straightforward: pick one handle. Key point prediction can already tell where are the handles, Is it possible to integrate this information into the pick-and-place policy.

**Quality Of The Limitations Section:**

Limitations are addressed clearly

**Reviewer Expertise:**

4: The reviewer is confident but not absolutely certain that the evaluation is correct

**Robotics Focus:**

Sufficient demonstration on hardware

**Strengths And Weaknesses:**

Strengths:
- Knotting plastic bags is a novel task and it is very common in our daily life.
- Although there are many modules in the proposed method (IIM), each of them is carefully chosen and the whole pipeline is well designed. I really appreciate this system design.
- Keypoint is a good choice for the intermediate representation. It is low-dimension, explicit, and able to contain almost all the necessary information to describe the bag state.
- A lot of real-world experiments are executed to demonstrate the performance of this method.

Weakness:
- It will be better to describe each module in more detail. For example, what's the training supervision for spatial action map. The demonstration can only support positive data. Then how about negative data, all remaining pixels are negative?


**Summary Of Recommendation:**

Knotting plastic bags is challenging enough and very novel in robot learning community.
Although no novel submodule are proposed in this paper, It takes advantage of each submodule and uses the appropriate intermediate representation (key points) to build a system very carefully.
FInally, A lot of real-world experiments are executed.

---

> ### Author Response · Authors · 2022-08-19
> **Response to Reviewer 5bQ2**
>
> Dear reviewer 5bQ2,
>
> We sincerely appreciate your valuable and insightful comments. We address your questions in detail below.
>
> ### Q1: How to train the spatial action map using demonstrations? How about negative data?
>
> A1: Our spatial action maps model is trained in a common way as in previous works. That is, given the ground truth grasping locations on training images (from demonstrations), we train a fully convolutional network to generate the heatmaps that represent the affordance of grasping points.
>
> We are sorry that we are not quite sure about your meaning of “negative data”. Do you mean that there are some non-optimal data in demonstrations? In this work, we just use all demonstrations for training because they all achieve successful results.
>
> ### Q2: A scaled and rotated spatial action map may be used in your work.
>
> A2: We have used rotated spatial action maps in our method. Scaling is useful in FlingBot to predict the dual-arm picking positions since the grasping distance between two arms is an essential element that influences the final manipulation results, but in our task, this is not very important. We aim to grasp the handles of plastic bags, and the distance between two handles is not quite important, as long as they are far enough from each other to avoid collisions of robot arms. Thus we think scaling is not that helpful in our task. Previous works like TossingBot didn’t use scaling either.
>
> ### Q3: Is it possible to integrate keypoint detection information into the pick-and-place policy?
>
> A3: During the pick-and-place process, the plastic bag is still in self-occlusion and partial observable situation, and the keypoint detection results are still very unstable, as shown in the supplementary video. So if one wants to integrate keypoint detection information into the pick-and-place policy, he or she must come up with a method to deal with unreliable keypoint detection.

---

### Official Review · Reviewer_NmgN · 2022-08-01

**Originality:** Good
**Technical Quality:** Very Good
**Clarity Of Presentation:** Very Good
**Impact:** 3

**Recommendation:**

Weak Accept: I recommend accepting the paper, but will not argue for my recommendation if the majority of other reviewers have a different opinion.

**Summary:**

The paper describes a learning method for knotting plastic bags with a dual arm robot. The proposed iterative interactive modeling (IIM) method will first adjust the bad to a predefined-pose for better keypoints detection then perform few primitives to knot it. The paper combines GNN, 2D keypoints detection RLE model and motion primitives together to accomplish the knot task with 100 demonstrations for each plastic bag. Also, the work provides a 2D plastic bad keypoints dataset as well.

**Issues:**

N/A

**Quality Of The Limitations Section:**

Limitations are addressed clearly

**Reviewer Expertise:**

5: The reviewer is absolutely certain that the evaluation is correct and very familiar with the relevant literature

**Robotics Focus:**

Sufficient demonstration on hardware

**Strengths And Weaknesses:**

Strengths:
1. The paper addresses a very interesting and novel task -- plastic bag knotting. And the paper tries to shape the plastic bag to equilibrium pose for better perception and planning input instead of directly knotting the plastic bag from random initial configurations.
2. The presentation for method is clear and straight forward. Figures clearly express the algorithm step by step.
3. There are sufficient quantitative results, baseline comparison and real world execution videos look promising.

Weaknesses:
1. It seems to me the visual grasping module assumes the robot conducts 4DOF top-down picking and placing actions to shape the plastic bag. And the heatmap is predicted from top down images. However, what if the handle needs to be grasped from side? Similar to the Transporter Networks, is that possible to extend this module to 6DOF grasping?
2. The performance on new grey bag drops a lot compared to the training result. Could you explain more about it? To me, the train and test objects are nearly the same except a little bit size, color different. However adding data augmentation to the CNN layers inside your modules should be able to handle those tiny changes and generalize well. It would be better if you could show the train/test loss curve to show the model is not overfitting.
3. As you mentioned in the limitation part, the method assumes the bag needs to be hard enough to be able to stand still in the first phase.


**Summary Of Recommendation:**

The paper shows a novel solution of knotting bag task by first shaping the bag's pose then executing motion primitives.
It would be better if you could show how you collect the expert data in a video and some real execution heatmaps generated from the visual grasping module. As to the crappy depth issue from colmap, maybe structure light sensor is a way to get the better depth for the z location in picking.
Though there are some limitations, concerns for the paper, I'd like to accept this paper. And I am looking forward the answers I described above during the rebuttal period and happy to discuss.

---

> ### Author Response · Authors · 2022-08-19
> **Response to Reviewer NmgN**
>
> Dear reviewer NmgN,
>
> We sincerely appreciate your valuable and insightful comments. We address your questions in detail below.
>
> ### Q1: Is that possible to extend the visual grasping module to 6DOF grasping?
>
> A1: Extending the visual grasping module to 6DOF may sound attractive, but we do not do that for two reasons.
>
> Firstly, extending it to 6DOF is almost as difficult as knotting plastic bags from randomly dropped initial configurations, since if we can reliably grasp the handle in a 6DOF way, we can directly tie the knot after grasping. The core idea of this paper is to decompose the original problem into two simpler sub-problems, and using 4DOF top-down picking and placing is one of the simple sub-problems.
>
> Secondly, in our experiments, there are no such kinds of handles that need to be grasped from the side. In our attached video, even if the handle is pointing to the side (rather than pointing upward), grasping it from the top is feasible, and the handle will be shaped upward after being iteratively grasped several times. That is one of the strengths of our proposed iterative interactive modeling method.
>
> ### Q2: Could you explain more about why the performance on the new grey bag drops a lot compared to the training result?
>
> A2: Thanks for your question. In fact, we say that the performance drops on grey bags are not from the visual perception part, but from the physical properties of the grey plastic bags. The grey plastic bag is much softer than the three learned ones, which is harder for it to stand steadily by itself and is more susceptible to indoor airflow. Besides, the grey one is the smallest plastic bag in all plastic bags, which makes the filler easier to fall out when the bag drops toward the ground, as shown in Figure 8(c).
>
> ### Q3: The method assumes the bag needs to be hard enough to be able to stand still in the first phase.
>
> A3: Yes, that is indeed a limitation of our method, but we say both hard and soft plastic bags can be used in our task, not just hard ones as you saying. In our limitation part, we said “For very soft/hard plastic bags … our would not be effective”. In fact, we found most plastic bags can satisfy our requirements since as long as there is no indoor wind, most plastic bags can stay in an equilibrium state by themselves.

---

### Meta-Review · Area_Chair_hjgF · 2022-08-11

**Recommendation:** Accept (Poster)
**Confidence:** 4

**Metareview:**

Reviewers felt positively that the task is interesting and novel.  The proposed method is reasonable, well-designed, and presented clearly.  The results appear to be very promising, and reviewers were happy to see so many real robot experiments.

Reviewers felt that their major concerns have been resolved during the rebuttal period.

Reviewers also noted a few limitations of the paper: the proposed method has strong assumptions and restrictions on the types of bags that can be used. Reviewers also requested to see a video showing the expert data collection as well as some real execution heatmaps generated from the visual grasping module.

Although the reviewers felt that the paper should likely be presented as a poster and not as an oral presentation, there was also some feeling that perhaps the paper could be nominated as “Best Systems Paper” due to the impressive results of the overall proposed system on a very challenging task.

**Best Paper Nomination:**

Yes